

# The biomarkers of key miRNAs and target genes associated with acute myocardial infarction

Qi Wang[1], Bingyan Liu[2,3], Yuanyong Wang[4], Baochen Bai[1], Tao Yu[3] and Xian–ming Chu[1,5]

[1] Department of Cardiology, The Affiliated hospital of Qingdao University, Qingdao, China
[2] School of Basic Medicine, Qingdao University, Qingdao, China
[3] Institute for Translational Medicine, Qingdao University, Qingdao, China
[4] Department of Thoracic Surgery, Affiliated Hospital of Qingdao University, Qingdao, China
[5] Department of Cardiology, The Affiliated Cardiovascular Hospital of Qingdao University, Qingdao, China

Corresponding authors
Tao Yu, yutao0112@qdu.edu.cn
Xian–ming Chu,
18661801698@163.com

## ABSTRACT

**Background**. Acute myocardial infarction (AMI) is considered one of the most prominent causes of death from cardiovascular disease worldwide. Knowledge of the molecular mechanisms underlying AMI remains limited. Accurate biomarkers are needed to predict the risk of AMI and would be beneficial for managing the incidence rate. The gold standard for the diagnosis of AMI, the cardiac troponin T (cTnT) assay, requires serial testing, and the timing of measurement with respect to symptoms affects the results. As attractive candidate diagnostic biomarkers in AMI, circulating microRNAs (miRNAs) are easily detectable, generally stable and tissue specific.
**Methods**. The Gene Expression Omnibus (GEO) database was used to compare miRNA expression between AMI and control samples, and the interactions between miRNAs and mRNAs were analysed for expression and function. Furthermore, a protein-protein interaction (PPI) network was constructed. The miRNAs identified in the bioinformatic analysis were verified by RT-qPCR in an H9C2 cell line. The miRNAs in plasma samples from patients with AMI ($n = 11$) and healthy controls ($n = 11$) were used to construct receiver operating characteristic (ROC) curves to evaluate the clinical prognostic value of the identified miRNAs.
**Results**. We identified eight novel miRNAs as potential candidate diagnostic biomarkers for patients with AMI. In addition, the predicted target genes provide insight into the molecular mechanisms underlying AMI.

## INTRODUCTION

Acute myocardial infarction (AMI) is the most common cardiac event worldwide and among cardiovascular diseases (CVDs) is a leading threat to human health (*Guo et al., 2019*). AMI is caused by acute coronary syndrome (ACS), which is induced by plaque ulceration or intravascular thrombosis and thrombotic material after rupture (*Li, Zhou & Huang, 2017*). Early diagnosis and interventional therapy are important to minimize

the damage to cardiac muscle and have the potential to significantly reduce mortality and improve prognosis (*Braunwald, 2012*). Although the cardiac troponin T (cTnT) assay, the gold standard for diagnosis of AMI, has facilitated the diagnosis of AMI and contributed to lower mortality, it can lead to false positives in patients with chronic but stable coronary artery disease or healthy controls (*Braunwald, 2012*). Therefore, novel biomarkers with high sensitivity and specificity are urgently needed to allow the early diagnosis of AMI and thereby improve clinical outcomes.

microRNAs (miRNAs), which are RNAs containing approximately 20 to 24 nucleotides, do not have the potential to encode proteins but can negatively regulate genes (*Cheng et al., 2019*). miRNAs restrain protein translation or mRNA degradation by binding to the 3′UTRs of messenger RNAs (mRNAs) (*Fasanaro et al., 2010*). Accumulating studies have revealed that miRNAs are involved in multifarious biological functions, including cell proliferation, apoptosis and inflammation, and exhibit strong correlations with mechanisms of disease, especially in cardiovascular disease (*Feinberg & Moore, 2016*). miRNAs have been identified as biomarkers of pathological events during the process of AMI (*Boon & Dimmeler, 2015*). In particular, the knockdown of miR-155 inhibits cardiomyocyte apoptosis in AMI-induced mice, and miR-155 is upregulated by negatively regulating the RNA-binding protein Quaking (QKI) (*Guo & Liu, 2019*). *Cai & Li (2019)* found that miR-29b-3p overexpression could protect cardiomyocytes against hypoxia-induced injury by negatively regulating the level of TRAF5, which suggests a potential therapeutic method for AMI. Circulating miRNAs are easily detectable, relatively stable and tissue specific, making them attractive candidate biomarkers (*Wang et al., 2010*). Enhancing our understanding of the relationships between miRNAs and target genes can help reveal detailed mechanisms and identify novel biomarkers for AMI. In this study, we aimed to identify miRNAs with high clinical applicability for distinguishing patients with AMI from those without.

To that end, we identified circulating miRNAs that are differentially expressed (DE) in AMI by using integrated analysis. Gene expression profiles in AMI were acquired through the Gene Expression Omnibus (GEO) database. Then, a competitive endogenous RNA (ceRNA) network was constructed after a comprehensive analysis. Receiver operating characteristic (ROC) curve analysis was applied to analyse the diagnostic usefulness of the identified DE-miRNAs and genes. Finally, eight potential miRNAs were identified as significant predictors of AMI. Our study may be helpful for elucidating the mechanisms of AMI pathogenesis and identifying diagnostic biomarkers for AMI.

## MATERIAL AND METHODS

### Subjects

A total of 11 AMI patients and 11 healthy subjects were enrolled from Affiliated Hospital of Qingdao University between 2017 and 2018 (Table S2). All of the AMI patients had been diagnosed for the first time and undergone a primary percutaneous coronary intervention (PCI). The diagnosis for AMI was made based on the following criteria: (i) acute ischaemic chest pain within 24 h; (ii) electrocardiogram changes (pathological Q wave, ST-segment

elevation or depression) and (iii) increases in cardiac biomarkers. The exclusion criteria were selected due to their potential influence on miRNA expression and were as follows: previous history of cardiac disease, tumour, renal insufficiency, surgery within the six previous months, and anticoagulant therapy. The study was conducted in accordance with the Declaration of Helsinki. The ethical committee of Affiliated Hospital of Qingdao University approved the study numbered QYFYWZLL25621.

## Sample collection and RNA isolation

Blood samples were collected into EDTA tubes before coronary angiography and application of heparin. Serum was obtained after centrifugation at 3,000 g for 10 min at 4 °C to remove debris and stored in RNase-free tubes at −80 °C at Affiliated Hospital of Qingdao University until analysis. All participants were informed of the study details by the ethics committee of the hospital and provided written informed consent. Total RNA from the serum samples was extracted using TRIzol reagent (Sigma, St. Louis, MO, USA) following the manufacturer's instructions. For normalization, 25 fmol *Caenorhabditis elegans* miR-39 (cel-miR-39) (Qiagen, Valencia, CA) was added to each serum sample after the addition of TRIzol, following previous methods.

## Data sources

The data expression profiles of AMI were searched in the GEO database, and two independent datasets of research on AMI, GSE24591 and GSE31568, were included in our study. Using the genome-wide expression data of miRNAs obtained from the two selected independent cohorts, differential genes were screened according to the control group and AMI group of samples.

## Data preprocessing and identification of DEGs

GEO2R, which is an interactive web tool that allows comparisons between two groups of samples to analyse almost any GEO series, was used to confirm DEGs between the control group and AMI group. The limma R package was applied by GEO2R and served as the processor to handle the supplied processed data tables. DEGs between the control and AMI groups were screened out according to the criteria $p$ value less than .05 and absolute log fold change greater than 1.

## Analyses of miRNA-mRNA targets

Investigating the target genes of miRNAs is crucial for identifying the regulatory mechanisms and functions of miRNAs. Herein, we identified 8 DE-miRNAs and then predicted the targets of the DEGs by employing three miRNA-target tools: miRWalk V2.0 database, mirDIP, and miRTarBase. The miRNA targets were screened based on the overlapping results from the three websites. Then, the regulatory networks of the miRNA-mRNA pairs were extracted (based on an expression fold change >2.5 and an FDR <0.05) and visualized using Cytoscape software (*Smoot et al., 2011*).

**Table 1  The primer used in QPCR.**

| | |
|---|---|
| hsa-miR-545 | Forward: 5′-TCAGTAAATGTTTATTAGATGA-3′ |
| hsa-miR-139-3p | Forward: 5′-GGAGACGCGGCCCTGTTGGAG-3′ |
| hsa-miR-101 | Forward: 5′-CGGCGGTACAGTACTGTGATAA-3′ |
| hsa-miR-24-1* | Forward: 5′-TGCCTACTGAGCTGATATCAGT-3′ |
| hsa-miR-598 | Forward: 5′-CGTACGTCATCGTTGTCATCGTCA-3′ |
| hsa-miR-33a | Forward: 5′-AGCGTGCATTGTAGTTGCATTGCA-3′ |
| hsa-miR-142-3p | Forward: 5′-TGTAGTGTTTCCTACTTTATGGA -3′ |
| hsa-miR-34a | Forward: 5′- CGCGTGGCAGTGTCTTAGCT-3′ |

## Gene Ontology (GO) annotation and Kyoto Encyclopaedia of Genes and Genomes (KEGG) pathway enrichment analyses of the DEGs

GO annotation and KEGG pathway enrichment analyses of the DEGs were performed using the Database for Annotation, Visualization and Integrated Discovery (DAVID) (selected with enrichment significance evaluated at $p < .05$), which revealed the biological processes (BPs), cellular components (CCs), molecular functions (MFs) and pathways associated with the DE-miRNAs.

## Protein–protein interaction (PPI) network construction and hub gene identification

To gain insights into the interactions of the 591 target genes of the identified miRNAs, a PPI network was constructed and analysed with the STRING tool to reveal the molecular mechanisms underlying AMI. Target genes in the PPI network serve as nodes, the lines between two nodes denote associated interactions, and the strengthof the interaction is expressed by the colour of the line. The hub genes, which were defined as genes that play essential roles in the network, were distinguished according to the cutoff criteria of degree calculated by cytoHubba in Cytoscape. The corresponding interactions were visualized using Cytoscape software (http://cytoscape.org/) (*Su et al., 2014*).

## Cell culture and treatment

The H9C2 cell line was obtained from the Shanghai Institutes for Biological Sciences (Shanghai, China) and cultured in Dulbecco's modified Eagle's medium (DMEM) containing 10% FBS (ExCell Bio, Shanghai, China) and 1% antibiotics. The cells were cultivated in a humidified atmosphere with 5% CO2 at 37 °C. The cells were trypsinised to generate single cell suspensions at 80% confluency. The cells were treated with 2 μM doxorubicin (DOX) for 24 h.

### RT-qPCR analysis

A reverse transcription kit (Takara, Otsu, Japan) was applied to synthesize the cDNA. RT-qPCR was accomplished using SYBR Green PCR Master Mix (Yeasen, Shanghai, China). The forward primers arepresented in Table 1, and the same reverse primer with the sequence 5′-GTGCAGGGTCCGAGGT-3′ was used for all miRNAs.
The average expression levels of serum miRNAs were normalized against cel-miR-39 (Qiagen, Valencia, CA), and the expression of cell-derived miRNAs were normalized against U6 (Takara, Otsu, Japan).

Fold changes in miRNA expression were calculated using the $2^{-\Delta\Delta Ct}$ method for each sample in triplicate (*Chang, Chen & Yang, 2009*). Taking the calculation method of miRNA expression in plasma as an example, $\Delta\Delta Ct = [(Ct_{miRNA} - Ct_{cel-miRNA-39})_{diseased} - (Ct_{miRNA} - Ct_{cel-miRNA-39})_{control}]$. In brief, with this method, the Ct values from the target miRNA in both AMI and control group are adjusted in relation to the Ct of a normalizer RNA (cel-miR-39), which resulted in $\Delta$Ct. In order to compare diseased and control samples, we calculated $\Delta\Delta$Ct values, which allowed us to determine the magnitude of the difference in miRNA expression. To ensure consistent measurements throughout all assays, for each PCR amplification reaction, three independent RNA samples were loaded as internal controls.

### ROC curves

ROC curves were constructed to discriminate AMI patients from control subjects for the plasma miRNAs, and the areas under the ROC curves (AUCs) were analysed to assess the diagnostic accuracy of each identified miRNA. Herein, a normalized miRNA score was used to represent the expression level of the selected miRNA in the AMI group relative to that in the control group (*Goren et al., 2012*). In brief, we used miRNA scores, which were calculated by subtracting the normalized Ct from 40 and then adjusted by deducting the minimal score, leading to miRNA scores with a lower bound of 0. All statistical analyses were performed using SPSS 13.0 (Chicago, IL, USA).

### *Statistical analyses*

Student's $t$-test was carried out using GraphPad Prism 5 to compare test and control samples. For the analysis of clinical characteristics in AMI patients and control individuals, data were presented as means $\pm$ standard deviations (SD) for quantitative variables. Mean values of quantitative variables were evaluated by Student's $t$-test, or Mann–Whitney $U$-test when Student's $t$-test were not satisfied. $p < .05$ was considered to indicate a statistically significant difference. All statistical analyses were performed using SPSS 13.0 (Chicago, IL, USA).

## RESULTS

### Identification of DE-miRNAs

|Log2FC|>1 and $p$ value <.05 were considered as criteria to screen the DE-miRNAs. Among the selected GEO datasets, 27 DE-miRNAs, including 25 downregulated and 2 upregulated genes, were found in the GSE24591 profile (Fig. 1A), whereas 307 DE-miRNAs, including 132 upregulated and 175 downregulated genes, were found in the GSE31568 profile (Fig. 1B). The candidate DE-miRNAs generated by the two datasets were intersected using a Venn diagram (Fig. 1C). All intersecting DE-miRNAs are shown in Table 2 and included hsa-miR-545, hsa-miR-139-3p, hsa-miR-101, hsa-miR-24-1, hsa-miR-598, hsa-miR-33a, hsa-miR-142-3p, and hsa-miR-34a.

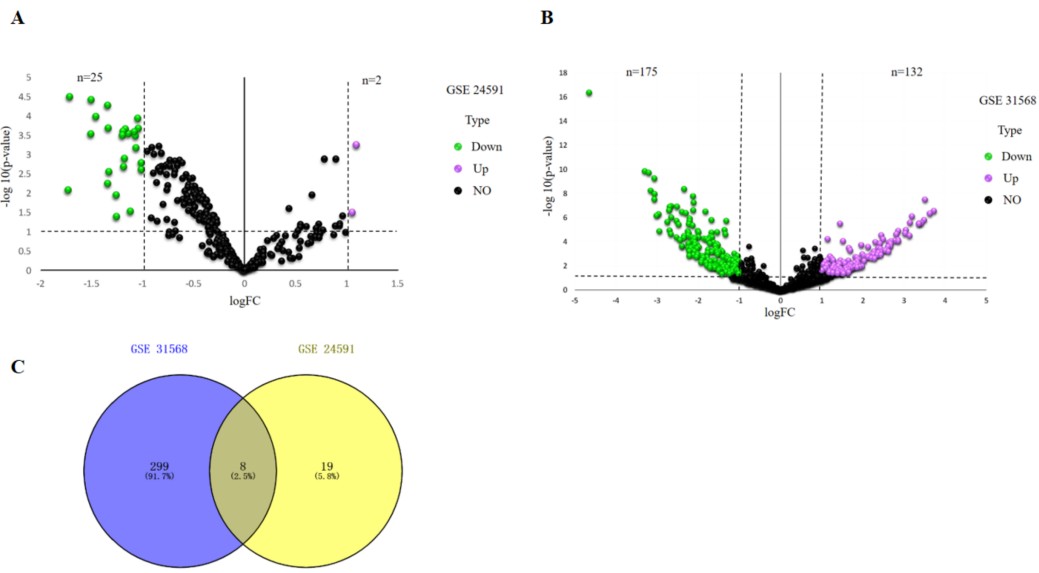

**Figure 1 Identification of differentially expressed miRNAs analysis.** (A) Volcano plot of differentially expressed miRNAs in GSE24591. The red dot represents upregulated miRNAs and the green dot represents downregulated miRNAs. (B) Volcano plot of differentially expressed miRNAs in GSE31568. The red dot represents upregulated miRNAs and green dot represents downregulated miRNAs. miRNAs, microRNAs. (C) A Venn-diagram between GSE24591 and GSE31568. The coincident part represents the differentially expressed genes shared by the two series, accounting for a total of eight.

**Table 2 The DE-miRNAs.**

| Symbol | *P* Value | logFC | Up/Down |
|---|---|---|---|
| hsa-miR-545 | 0.00038 | −1.50467 | Down |
| hsa-miR-139-3p | 0.0005900000 | −1.53651 | Down |
| hsa-miR-101 | 0.0080000000 | 1.04767 | Up |
| hsa-miR-24-1* | 0.0001180000 | −2.08228 | Down |
| hsa-miR-598 | 0.0036000000 | −1.14074 | Down |
| hsa-miR-33a | 0.0002760000 | −1.98301 | Down |
| hsa-miR-142-3p | 0.0002622 | −1.07909 | DOWN |
| hsa-miR-34a | 0.0031100000 | 1.90204 | Up |

## miRNA-target gene interactions

Following data preprocessing and analysis of the three databases, an overlap of 591 gene pairs from eight DE-miRNAs was obtained among the databases. These overlapping pairs were used to predict the target genes that interact with the miRNAs. The predictions were verified by more than four algorithms, including miRDB, RNA22, RNAhybrid, and TargetScan. The network of miRNA-mRNA interactions was visualized in Cytoscape, as shown in Fig. 2, and the target genes are listed in Table 3.

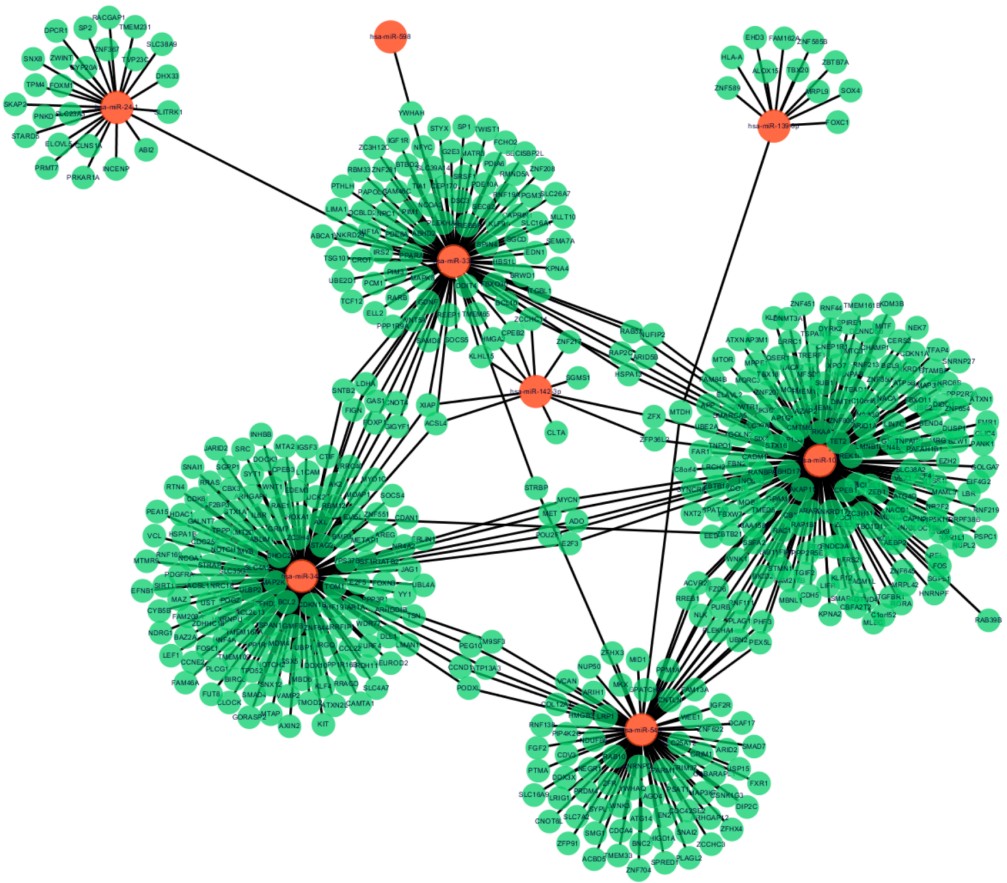

**Figure 2** **MiRNA-target gene interactions.** Interaction networks of miRNA and target DEGs in AMI. The red dot represents miRNAs and the green dot represents target mRNAs.

## Enrichment analyses of the target genes

To investigate the functions of the target genes, GO annotation and KEGG pathway analyses of the interacting 591 genes from GSE24591 and GSE31568 were performed utilizing the DAVID online tool. The top 10 GO and KEGG items, including the BPs, CCs, MFs and KEGG pathways that were significantly enriched, are listed in Figs. 3A–3D. The significantly enriched entries for BPs were positive regulation of transcription from the RNA polymerase II promoter, transcription, and DNA-templated and negative regulation of transcription from the RNA polymerase II promoter (Fig. 3A). Furthermore, the nucleus, cytoplasm, and nucleoplasm accounted for the majority of the CC terms (Fig. 3B). The most enriched MFs were functions in metal ion binding, zinc ion binding and poly (A) RNA binding (Fig. 3C). In the MF category, the top 10 most highly regulated DE-miRNAs were significantly enriched in the pathways of cancer and the PI3K-Akt pathway. Intriguingly, the enrichment in adrenergic signalling in cardiomyocytes was found to be closely related to AMI (Fig. 3D).

Peerj

**Table 3  The miRNA-mRNA network.**

| Symbol | Up/Down | Count | Target mRNA |
|--------|---------|-------|-------------|
| hsa-miR-545 | Down | 85 | USP15, WEE1, DDX3X, CSNK1G3, LRP1, FGF2, CRIM1, FAM13A, ZFHX4, PEX5L, RAB10, AGO4, PLAG1, MAP3K7, HIGD1A, ZCCHC3, CNOT6L, PLEKHA1, PIP4K2C, NLK, ARID2, PSAT1, RNF138, PTMA, BNC2, FZD6, SLC25A12, SYPL1, COL12A1, LRIG1, NUP50, ARIH1, RNF111, CDC42SE2, TRIM37, GPATCH8, ZFR, PURB, ARHGAP12, SNAI2, PEG10, CCND1, YWHAQ, WNK3, PRDM4, HMGB3, FXR1, PLAGL2, SMAD7, ACVR2B, CDV3, RREB1, DIP2C, GABARAPL1, UBN2, CDCA4, MTDH, ATG14, TM9SF3, SLC7A2, HNRNPDL, DCAF17, SLC16A9, SPRED1, TMEM33, ZFHX3, ZNF704, VCAN, NDUFB6, MID1, PARM1, CNTLN, MKX, SMG1, PHF3, IGF2R, ZNF622, PODXL, PPM1A, NEGR1, ATP13A3, ACBD5, STRBP, EN2, ZFP91 |
| hsa-miR-139-3p | Down | 12 | MTDH, SOX4, ZBTB7A, EHD3, ALOX15, ZNF585B, FAM162A, ZNF589, TBX20, HLA-A, FOXC1, MRPL9 |
| hsa-miR-101 | Up | 205 | DUSP1, EZH2, FBN2, ATXN1, ARID1A, RAP1B, MYCN, TGFBR1, AEBP2, BICD2, FOS, BCL9, MBNL1, RAB5A, ANKRD17, ZNF207, RANBP9, RAP2C, MOB4, NLK, DNMT3A, ZCCHC2, FNDC3A, NACA, PTGS2, TNPO1, PAFAH1B1, MITF, RNF111, CBFA2T2, SMARCD1, ZBTB18, MAP3K4, SOX9, DYRK2, SMARCA5, LCOR, ZNF654, LMNB1, SUB1, HNRNPF, UBE2D3, ICK, MBNL2, SIX4, OTUD4, INO80D, ZEB2, APP, ABHD17C, MRGBP, ARID5B, CADM1, RREB1, MET, CDH5, STMN1, MFSD6, TSPAN12, TMEM161B, TET2, PURB, SYNCRIP, PPP2R2A, UBE2A, ZEB1, AP1G1, NR2F2, PPP2R5E, FMR1, TGIF2, ZFP36L2, ANKRD11, LIFR, PHF3, CERS2, NEK7, MPPE1, ZFX, PRKAA1, TNRC6B, GNB1, BZW1, TMED5, UBN2, CPEB1, DDIT4, FZD6, FBXW7, KLF12, LRCH2, ZNF451, EED, HNRNPAB, PIP5K1C, RORA, EIF4G2, SLC38A2, ATXN1L, RNF219, C1orf52, BCL2L11, NAP1L1, C8orf4, KDM6B, ZC3H11A, DIDO1, ZBTB21, KDM3B, MLEC, STAMBP, MTSS1L, ARAP2, POU2F1, ACVR2B, BEND4, PIK3C2B, NUFIP2, FAM84B, C10orf88, SPATA2, NUPL2, MAML3, PSPC1, SGPL1, KLF6, LRRC1, RAC1, TMEM170B, RAB39B, TMEM68, LBR, PLEKHA1, AKAP11, HSPA13, MCL1, AFF4, SACM1L, ZNF800, AP1S3, CAPN2, FRS2, SREK1IP1, MRPL42, FAR1, TRERF1, RNF213, WWTR1, NACA2, SLC39A6, WNK1, TFAP4, DAZAP2, CNEP1R1, CBX4, SSFA2, SPIRE1, GOLGA7, ATG4D, MORC3, TGFBR3, SNRNP27, ADO, TGOLN2, LIN7C, MNX1, PANK1, GPAM, MTOR, NAA30, TMTC3, TBC1D12, PRPF38B, BLOC1S6, ELAVL2, KIAA1586, TBX18, DENND5B, TNFAIP1, KPNA2, NXT2, RAB11FIP1, N4BP1, PEX5L, CLIC4, VEZT, NACC1, AP3M1, FBXO11, E2F3, TEAD1, CDKN1A, ZNF350, PLAG1, ZNF645, REL, CMTM6, STX16, XPO7, CHAMP1, RNF44, DIMT1, QSER1, FAM217B, ATP5B |

Wang et al. (2020), *PeerJ*, DOI 10.7717/peerj.9129

**Table 3** (*continued*)

| Symbol | Up/Down | Count | Target mRNA |
|---|---|---|---|
| hsa-miR-24-1 | Down | 25 | SLITRK1, FOXM1, CYP20A1, DPCR1, CLNS1A, RACGAP1, ZWINT, ELOVL5, DHX33, PRMT7, PRKAR1A, ZNF367, TVP23C, DDIT4, TPM4, SLC23A3, SKAP2, TMEM231, STARD5, SLC38A9, SP2, INCENP, SNX8, PNKD, ABI2 |
| hsa-miR-598 | Down | 1 | YWHAH |
| hsa-miR-33a | Down | 93 | KPNA4, ZNF281, NPC1, ABCA1, YWHAH, HMGA2, CROT, ARID5B, PIM1, ABHD2, IRS2, SLC26A7, RMND5A, GAS1, PAPOLG, SGCD, STYX, ZC3H12C, CPEB2, ANKRD29, BTBD2, FAM46C, NUFIP2, PTHLH, RAB5A, DSC3, PIM3, FOXP1, SLC16A1, PPARA, TSG101, PCM1, LDHA, ZCCHC14, TWIST1, PDE8A, SP1, FCHO2, REEP1, SOCS5, RBM33, SEMA7A, STRBP, HBS1L, PPP1R9A, PGM3, UBE2D1, SRSF1, MLLT10, RNF19A, LIMA1, XIAP, GDNF, SLC39A14, NCOA3, SAMD8, CREBBP, NFYC, RAP2C, SEC62, SNTB2, SECISBP2L, SPIN4, TCF12, HIF1A, GIGYF1, CNOT4, MAPK8, TIA1, PDIA6, FIGN, KLHL15, ITGBL1, BCL10, PLEKHA8, DCBLD2, ACSL4, TMEM65, EDN1, FBXO30, ELL2, G2E3, ZNF208, RARB, CAPRIN1, MATR3, PDE10A, BRWD1, IGF1R, KLF9, HSPA13, WNT5A, CEP170 |
| hsa-miR-142-3p | Up | 11 | CLTA, ZNF217, HMGA2, CPEB2, ZCCHC14, SGMS1, ACSL4, ZFX, XIAP, ZFP36L2, KLHL15 |
| hsa-miR-34a | Up | 159 | SYT1, NOTCH1, DLL1, PDGFRA, SATB2, E2F5, FUT8, LEF1, TPD52, FOXP1, UBP1, E2F3, JAG1, POGZ, MET, GALNT7, FOXN3, ZNF644, ACSL1, BCL2, NR4A2, VAMP2, ACSL4, SGPP1, MYCN, RRAS, PEA15, KLF4, CCNE2, MAP2K1, AXL, EVI5L, SAR1A, CAMTA1, YY1, SIRT1, LDHA, TMEM109, SLC4A7, BAZ2A, MTA2, CSF1R, FOSL1, ARHGAP1, GMFB, BMP3, INHBB, CCND1, MTMR9, NOTCH2, GAS1, PODXL, SNTB2, VPS37B, MDM4, ZDHHC16, PPP1R16B, GRM7, CPEB3, CDK6, IL6R, NCOA1, HSPA1B, TSN, SURF4, FAM46A, RDH11, LRRC40, CNOT4, VCL, PPP1R10, METAP1, PEG10, HOXA13, EFNB1, STX1A, ADO, STRAP, CLOCK, LMAN1, SMAD4, SOCS4, AREG, PLCG1, DOCK3, SLC35G2, POU2F1, PHF19, TM9SF3, CCL22, HNF4A, RAE1, EDEM3, MYB, CDC25A, TNRC18, PPP3R1, TPPP, SHOC2, TOM1, WNT1, BCL2L13, IGF2BP3, MOAP1, GORASP2, MYO1C, SRC, KIT, KMT2D, CBX3, UBL4A, GIGYF1, CYB5B, MBD6, HDAC1, SNAI1, ZNF551, SSX5, ARHGDIB, ERLIN1, FIGN, TSPAN14, ZC3H4, ULBP2, ATP13A3, CDKN1B, STAG2, NEUROD2, AXIN2, RTN4, RRAGD, FAM208A, MAZ, ATXN2L, ABLIM1, IGSF3, UST, CDAN1, JARID2, WDR77, LRRFIP1, SNX12, CTIF, NDRG1, TMOD2, UCK2, HNRNPU, BIRC5, MTAP, RBM12, TMEM167A, XIAP, SLC4A2, AK2, EFHD2, RNF169, IRGQ, DDX10, L1CAM |

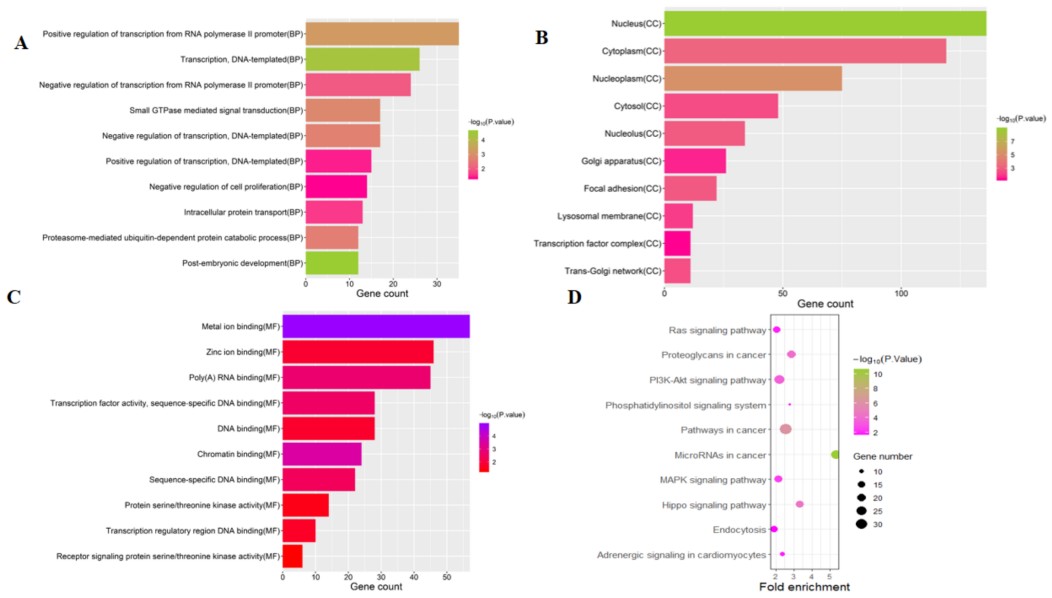

**Figure 3  Top 10 significant enrichment GO and KEGG terms of DEGs.** (A) BP: biological process; (B) CC: cellular component; (C) MF: molecular function; (D) KEGG: signaling pathway.

## PPI network

To distinguish the connections among the 591 target genes, we mapped the PPIs using the logical data originating from the STRING database (http://strin g.embl.de/). With degree as the criterion, the top 100 linked DE-miRNAs were identified, as shown in Fig. 4. The network is composed of 100 nodes and 700 edges and has an average local clustering coefficient of .467. The top 10 genes with a high-ranking degree are labelled in purple and associated with much larger circles; all the edges are distinguished based on connection score (Fig. 4).

## Biological analysis of the hub genes

Highly connected proteins in a network are master keys of regulation and are defined as hub proteins (*Yu et al., 2017*). The hub proteins in the present study included CTNNB1, CCND1, NOTCH1, EZH2, MTOR, BTRC, RAC1, CDKN1A, CDKN1B, and MAP2K1. They were identified by evaluating degree with the Biological Networks Gene Ontology tool (BiNGO) plugin of Cytoscape, which considered the top ten closely related interactions (Table 4); these involved 10 nodes and 35 edges (Fig. 5A). Additionally, KEGG analysis was performed on the potential hub genes (Fig. 5B), and the top 10 enrichment pathways were identified (Fig. 5C).

## Validation of the identified miRNAs

The expression levels of the identified miRNAs were quantified by RT-qPCR in H9C2 cells treated with DOX to verify the results of the bioinformatic analyses. Emerging studies have illuminated the role of cardiomyocyte apoptosis in DOX-induced myocardial damage, which is similar to the proceeding of AMI (*Catanzaro et al., 2019*). Based on our previous

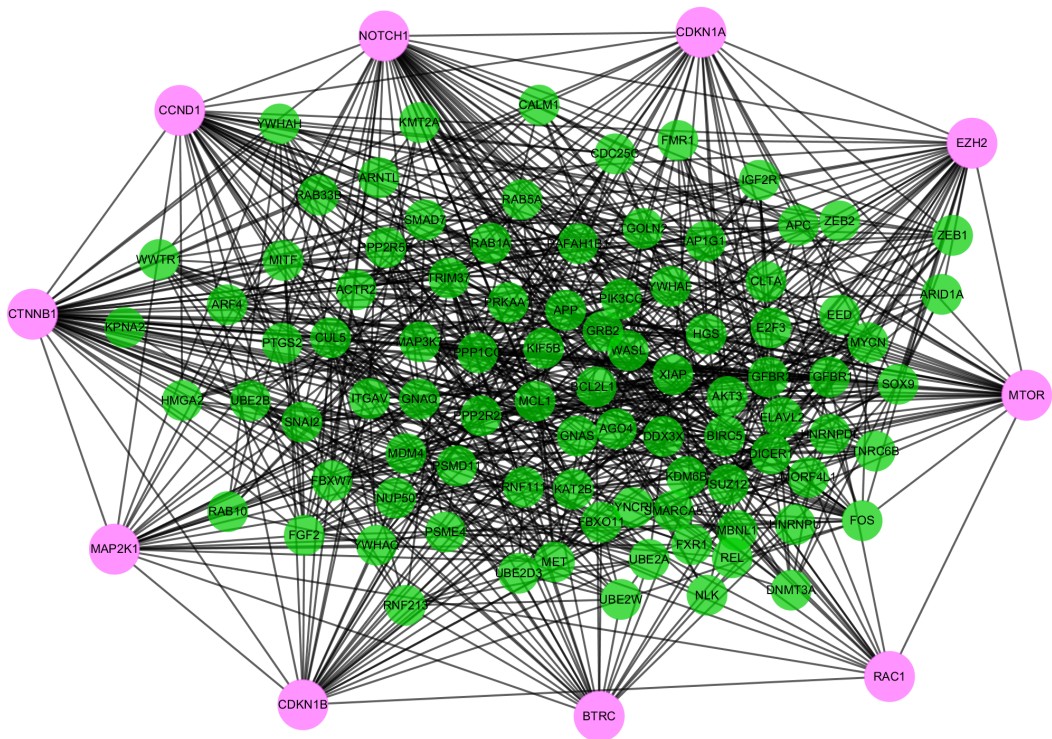

**Figure 4** **The PPI networks of top 100 DEGs.** All the circles are proteins encoded by top 100 DEGs. The red colors represent the 10 highest degree genes and the circles with green represent the remaining genes. Edges are distinguished using the color shading from white to yellow.

**Table 4  Top 10 genes in network ranked by degree method.**

| Rank | Symbol | Score |
|---|---|---|
| 1 | CTNNB1 | 50 |
| 2 | NOTCH1 | 47 |
| 3 | CCND1 | 41 |
| 4 | EZH2 | 36 |
| 5 | MTOR | 33 |
| 6 | CDKN1B | 31 |
| 7 | CDKN1A | 29 |
| 8 | MAP2K1 | 27 |
| 9 | BTRC | 26 |
| 10 | RAC1 | 24 |

study, we treated H9C2 cells with 2 μM DOX. As shown in Fig. 6, miR-34a, miR-101 and miR-598 were upregulated, and miR-24-1, miR-33a, miR-139-3p, miR-142-3p, and miR-545 were downregulated. Furthermore, the results of miR-24-1*, miR-33a, miR-34a, miR-101, miR-139-3p, and miR-545 in patients were consistent with the results obtained for the tissue cultures. However, the expression of miR-142-3p did not have significant change in blood samples and miR-598 was upregulated in blood but decreases in tissues

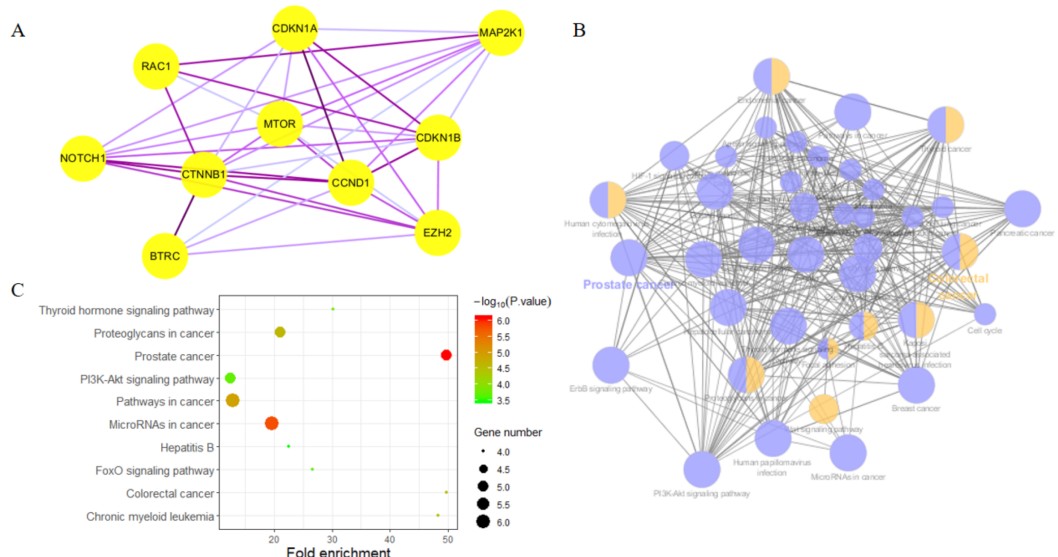

**Figure 5  Biological analysis of hub genes.** (A) The interaction of 10 hub genes; (B) the KEGG enrichment analysis by Cytoscape; (C) the top 10 KEGG enrichment analysis by R language.

(Fig. S1). To investigate the efficacy of DE-miRNAs as potential biomarkers of AMI, we performed ROC curve analysis of patients with AMI and patients without AMI. The expression levels of the DE-miRNAs were significantly different between AMI patients and control individuals (Fig. 7). AUC values were used to evaluate the potential of the DE-miRNAs as diagnostic markers. The AUC values of miR-24-1 and miR-545 were greater than .9, and these DE-miRNAs also had the highest accuracies. Moreover, all five miRNAs had high specificity with AUCs>.7 except for miR-142-3p that the accuracy is likely to take place when the AUC above .7 (Catanzaro et al., 2019). These results indicated that the predicted miRNAs, especially miR-24-1 and miR-545, have potential for clinical application.

### Relationships to conventional prognostic markers

To further evaluate the potential of circulating miRNAs as cardiac biomarkers, we tested whether the levels of identified miRNAs correlate with troponin T (TnT) level. miR-24-1 and miR-545 were strongly correlated with TnT ($r = -0.722$, $p < 1*10^{-3}$ and $r = -0.57$, $p = 0.006$). miR-101, miR-139-3p, and miR-598 remained correlated with TnT levels in AMI patients ($r = -0.444$, $p = 0.038$ for miR-101, $r = -0.425$, $p = 0.048$ for miR-139-3p and $r = -0.425$, $p = 0.048$ for miR-598). However, miR-33a, miR-34a, and miR-142-3p was not correlated with TnT which showed in Table S3. Combining ROC analysis results, we concluded that miR-24-1 might be the most potential biomarker in AMI.

## DISCUSSION

AMI, commonly referred to as acute heart attack, is generally acknowledged as the outcome of sudden ischaemia that results in insufficient blood supply and a subsequent imbalance

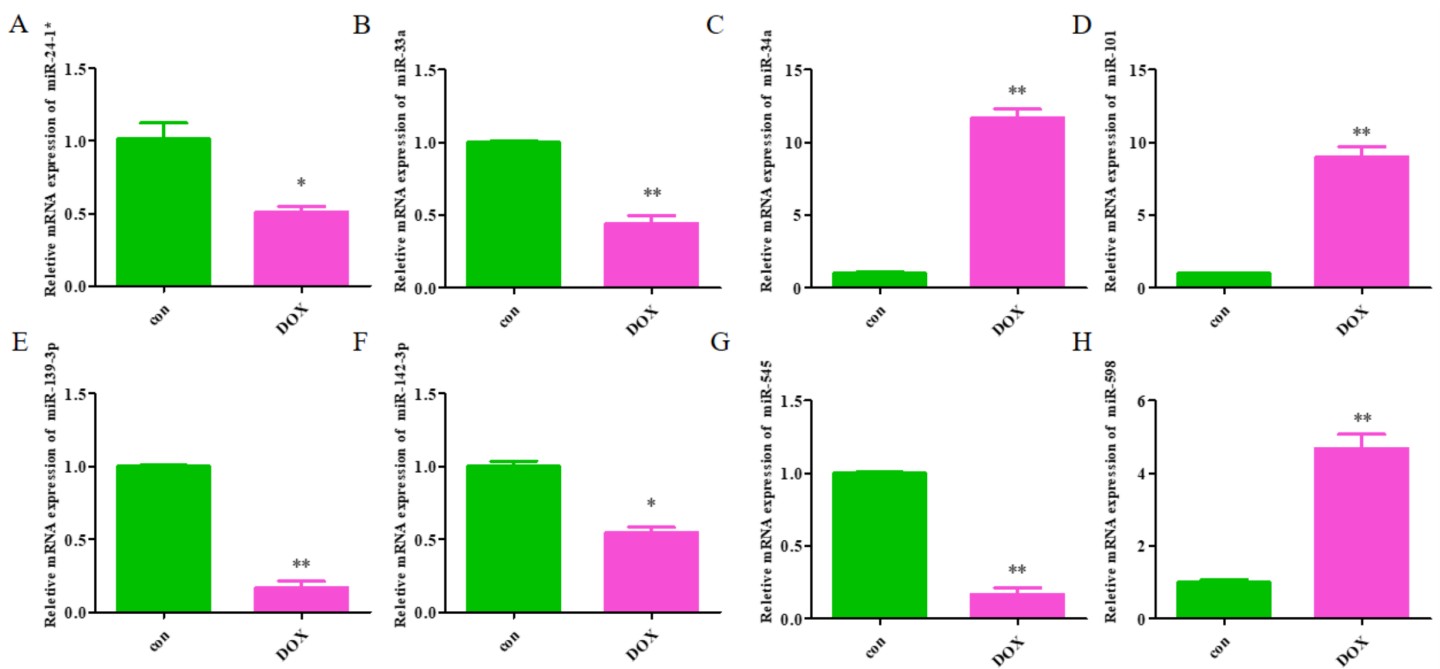

**Figure 6** **The relative expression of differentially expressed miRNA in H9C2.** (A) miR-24-1*; (B) miR-33a; (C) miR-34a; (D) miR-101; (E) miR-139-3p; (F) miR-142-3p; (G) miR-545; H, miR-598. The control group respects normal H9C2 and the DOX group respects the cell of H9C2 treated with DOX (2 µM).

between the supply and demand of oxygen induced by cardiomyocyte death (*Vogel et al., 2019*). AMI is a central contributor to the global disease burden, occurring in 4–10% of people under 45 years, with a massive number of patients still suffering recurrent cardiovascular events after treatment with medication or primary PCI (*Tan et al., 2016*). Previous studies have identified potential mechanisms and biomarkers for early diagnosis and treatment. Cardiac troponin (cTn) has served as the gold standard for AMI diagnosis and is routinely applied for patients with suspected ACS to rule-in or rule-out AMI (*Sandoval et al., 2017*). Nevertheless, with the advancing sensitivity of cTn assay, the assay has exceeded the ninety-ninth percentile for stable chronic conditions, weakening its specificity for the diagnosis of AMI (*Park et al., 2017*). This observation demonstrates that there is an urgent need for the identification of novel diagnostic markers and therapeutic targets with minimal risk of adverse effects and maximum sensitivity and specificity (*Cruz et al., 2019*). Various investigations have revealed that miRNAs can potentially predict CVDs by modulating the ceRNA network, thus providing a therapeutic option, especially in AMI (*Lucas, Bonauer & Dimmeler, 2018*). The downregulation of miR-155 expression restrains apoptosis and maintains a proliferative effect in cardiomyocytes by targeting QKI and can thereby serve as a therapeutic marker for MI. Accordingly, strategies for the diagnosis and treatment of AMI could be furnished by analysing correlative data in the GEO database (*Cruz et al., 2019*) and generating an AMI-associated miRNA-mRNA regulatory network for clinical applications regarding diagnosis, therapy, and prognosis.

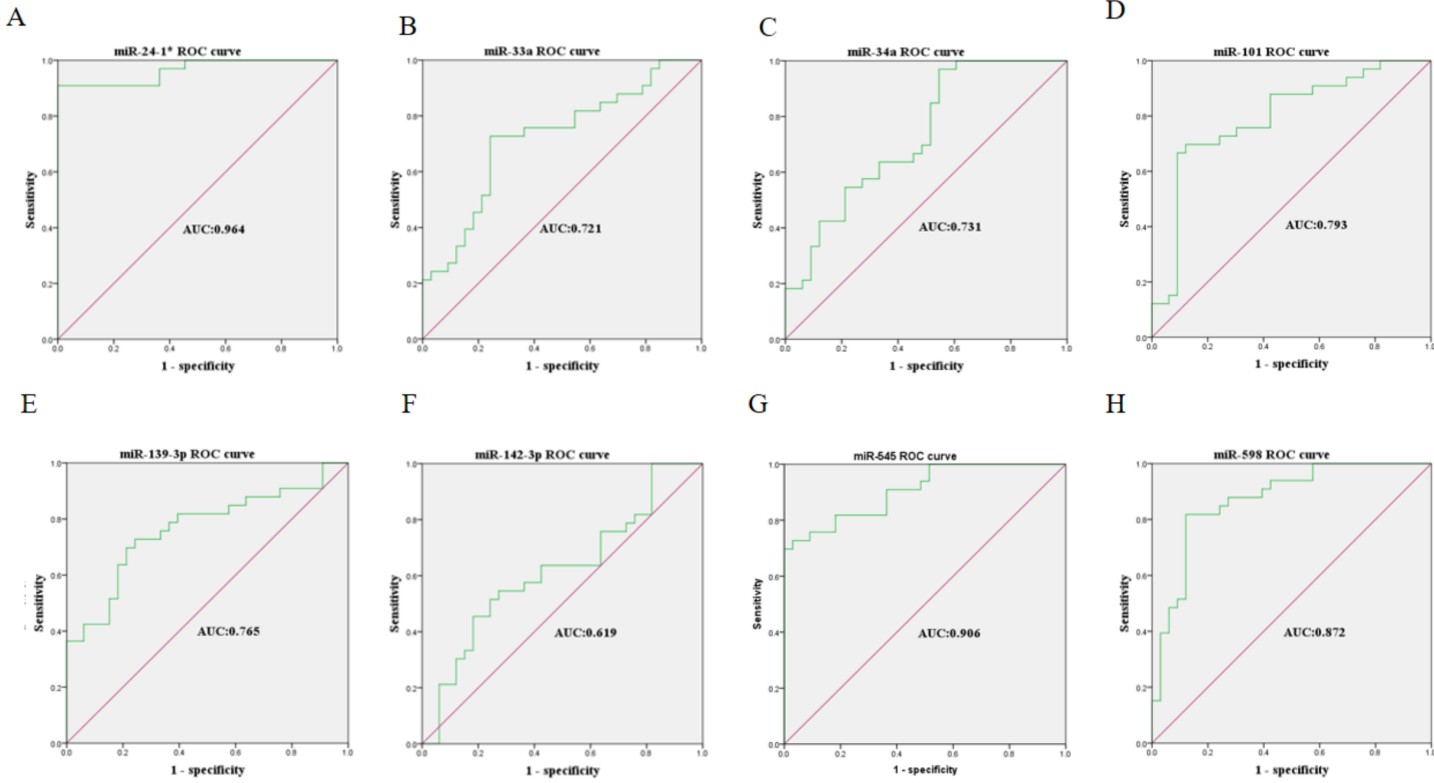

**Figure 7** **Receiver operating characteristic curves (ROC) of differentially expressed miRNA between AMI patients and healthy controls.** (A) miR-24-1*; (B) miR-33a; (C) miR-34a; (D) miR-101; (E) miR-139-3p; (F) miR-142-3p; (G) miR-545; (H) miR-598.

In this study, 27 DEGs in the GSE24591 dataset and 307 DEGs in the GSE31568 dataset were screened in AMI and control blood samples based on the differential analysis of GEO2R in the GEO database. Furthermore, 8 collective miRNAs (miR-545, miR-139-3p, miR-101, miR-24-1, miR-598, miR-33a, miR-142-3p, and miR-34a) were selected and identified as DE in AMI; there were few common DEGs because the two datasets were independent. An interaction network of miRNAs and mRNAs was constructed by using three websites, miRWalk V2.0, mirDIP, and miRTarBase, and more than 4 online prediction tools, including miRDB, RNA22, RNAhybrid, and TargetScan. In addition, GO and KEGG enrichment analyses of the mRNAs in the ceRNA network were performed. The PPI network, analysed via the STRING database and visualized by Cytoscape software, showed that 591 target proteins and 10 hub genes were significantly closely associated with the miRNAs. The expression validation and ROC analyses of these DE-miRNAs based on the RT-qPCR data supported the above results, which could quantify the diagnostic availability of the identified DE-miRNAs.

Moreover, among the 8 miRNAs that might exert effects on the development of AMI, miR-34a has been identified in mechanistic studies as a biomarker for AMI. It is highly expressed in adult mice after MI and associated with a thin wall of the left ventricle (LV) (*Qipshidze Kelm et al., 2018*). *Yang et al. (2015).* found that the suppression of miR-34a

facilitated cardiac function following MI partly by modulating the interrelated genes involved in cell proliferation and the cell cycle, including Bcl2, Cyclin D1, and Sirt1, which revealed the potential of miR-34a to boost endogenous repair/regeneration in the adult heart. Additionally, the remaining 7 miRNAs have been shown to regulate cardiac performance. miR-545, the negatively correlated target of HOTAIR, promotes cell apoptosis through the HOTAIR/miR-545/EGFR/MAPK axis (*Li et al., 2018*). miR-101 has been shown to mitigate the deterioration of cardiac function in post-MI rats (*Pan et al., 2012*), and it can protect cardiac fibroblasts from hypoxia-induced apoptosis by restraining the TGF-β signalling pathway, as shown by *Zhao et al. (2015)*. In contrast, miR-33 deteriorates myocardial fibrosis via the inhibition of MMP16 and the stimulation of p38 MAPK signalling (*Chen et al., 2018*). After I/R, the expression of miR-139-3p increases and is downregulated by Urocortin 1 (Ucn-1), and the overexpression of miR-139-3p promotes the expression of genes involved in cell death and apoptosis (*Díaz et al., 2017*). miR-24-1 was found to be significantly hypermethylated in ischaemic cardiomyopathy (ISCM) and dilated cardiomyopathy (DCM) and significantly reduced in an ISCM group (*Glezeva et al., 2019*). Additionally, miR-598 was identified as a significant predictor of heart failure (HF) in a dyspnoea cohort (*Ellis et al., 2013*), and miR-142-3p sponged by lncRNA TUG1 has been suggested to potentially alleviate myocardial injury (*Su et al., 2019*). Thus, more mechanistic research is needed to explore the potential functions of the identified miRNAs in AMI. We found that several hub genes and correlative mechanism pathways including Notch1, CTNNB1, RAC1, and MTOR had greater diagnostic potential for AMI. The Notch1 activation pathway manages cardiac AMPK signalling by interacting with LKB1 during myocardial infarction (*Yang et al., 2016*). Spermidine (SPD) has been suggested to be involved in the cardiac dysfunction induced by MI by promoting autophagy in the AMPK/mTOR pathway (*Yan et al., 2019*). The inhibition of Annexin A3 (ANXA3) has been reported to accelerate cardiomyocyte maintenance by activating PI3K/Akt signalling in rats with AMI (*Meng et al., 2019*). RAC1 has been shown to inhibit the death of cardiac myocytes stimulated by hypoxia and modify the phosphorylation levels of PI3K, AKT, MAPK and ERK, which are significant factors of MI (*Wang et al., 2017*). Overall, we inferred that the 4 hub genes might be regarded as diagnostic biomarkers and recovery monitors in AMI.

There are several limitations to our present study. The number of samples we obtained from GSE24591 and GSE31568 was small, generating some bias when analysing the DE-miRNAs, and more blood samples are needed for validation with RT-qPCR in further research. In addition, the functions and molecular mechanisms of genes are very complicated, and predictions based only on bioinformatics need cellular and animal experiments for verification.

## CONCLUSIONS

Based on GEO database analysis, bioinformatic analysis, and experimental verification, we not only identified eight significant DE-miRNAs in AMI but also detected 10 hub genes that may serve as potential biomarkers of AMI. Our findings might provide reliable

candidate biomarkers for the precise diagnosis and individualized treatment of AMI and the development of further clinical applications in AMI.

### Funding

This work was supported by National Natural Science Foundation of China (No. 31701208, 81870331), the Project of Shandong Province Higher Educational Science and Technology Program (No. J18KA285), and a project of Qingdao University Medical Department Clinical Medicine + X (No. 82911815)The funders had no role in study design, data collection and analysis, decision to publish, or preparation of the manuscript.

### Grant Disclosures

The following grant information was disclosed by the authors:
National Natural Science Foundation of China: 31701208, 81870331.
A Project of Shandong Province Higher Educational Science and Technology Program: J18KA285.
A Project of Qingdao University Medical Department Clinical Medicine + X: 82911815.

### Competing Interests

The authors declare there are no competing interests.

### Author Contributions

- Qi Wang conceived and designed the experiments, analyzed the data, prepared figures and/or tables, and approved the final draft.
- Bingyan Liu conceived and designed the experiments, prepared figures and/or tables, and approved the final draft.
- Yuanyong Wang and Baochen Bai performed the experiments, prepared figures and/or tables, and approved the final draft.
- Tao Yu and Xian–ming Chu analyzed the data, authored or reviewed drafts of the paper, and approved the final draft.

### Human Ethics

The following information was supplied relating to ethical approvals (i.e., approving body and any reference numbers):

Ethics Committee of the Affiliated Hospital of Qingdao University approved this study (Ethical Application Ref: QYFYWZLL25621).

### Data Availability

The raw measurements of Fig. 6 are available in the Supplementary Files.

### Supplemental Information

Supplemental information for this article can be found online at http://dx.doi.org/10.7717/peerj.9129#supplemental-information.

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
