# Peer review of "The biomarkers of key miRNAs and target genes associated with acute myocardial infarction"

_PeerJ, doi:10.7717/peerj.9129_

## Round 0.1 · original submission · Major Revisions

Your manuscript has been evualated by three Reviewers, who raised major criticisms to the rationale and experimental design of your study. Of note, the final comment of one Reviewer was rejection.

I think that PeerJ could consider a new version of your manuscript. However, I suggest you to consider very carefully each criticism raised by the Reviewers and, if you think that you cannot answer even at one issue, consider the opportunity of submitting your manuscript to another journal.

Reviewer 1 ·

Basic reporting

1. The text of the manuscript must be corrected by a native English speaker. The current title of the manuscript is not easily understandable.
2. Literature is relevant and well referenced.
3. Figures and Tables are relevant to the content of the article, and appropriately described and labeled.
4. Raw data is available in accordance with the Data Sharing policy.
5. Abbreviations, when first used, should be explained in the text as well as in a separate section.

Experimental design

1. The research topic of study is not novel. Numerous studies have been undertaken to identify and understand miRNA/mRNA biomarkers in acute myocardial infarction.
2. The aim of the study should be more emphasized in the text of the manuscript.
3. The number of analyzed samples acquired from the GEO database (normal and AMI groups) should be provided in the text of the manuscript.
4. The text of the paragraph “Analyses of miRNA-mRNA targets” should be more detailed.
5. Full names of the programs used for data preprocessing and expression data analysis should be provided.
6. The rationale for the selected dosage of doxorubicin should be clearly explained. Did the Authors test any other doses of doxorubicin?
7. More detailed demographic and clinical characteristics of the AMI patients could give valuable information.
8. What is the source of the investigated miRNA from the AMI patients?
9. U6 is not a suitable reference control for the quantification of circulating miRNAs. U6 is not miRNA, and does not reflect the biochemical character of miRNA molecules in terms of their transcription, processing and tissue-specific expression patterns. Were some other endogenous miRNA controls tested as a reference control?
10. The sequences of reverse primers for the verified miRNA should be provided.
11. More detailed description of the RT-qPCR statistical data analysis and ROC curves should be introduced into the text of manuscript.
12. It is not clear if the RT-qPCR experiments were performed using only H9C2 cell line or also with the AMI patients’ blood samples?
13. According to the MIQE Guidelines (Bustin el al., 2009), the abbreviation RT-qPCR (reverse transcription quantitative real-time PCR) should be used in the context of quantitative RT-PCR.
14. It is not clear what kind of data was used to construct the ROC curves.

Validity of the findings

1. All underlying data has been provided; it is robust, statistically sound, and controlled.
2. Conclusions are well stated, linked to the original research question and limited to supporting results. The limitations of the study are pointed out.

Reviewer 2 ·

Basic reporting

Some well known biomarkers are use for diagnosis of acute myocardial infarction, but identification of new biomarker can improve effective treatment for better prognosis of AMI patients. The aim of this study was to investigate miRNAs expression in AMI patients and identification of miRNAs as potential diagnostic biomarkers for AMI patients. To do this, the authors used microarrays results deposited in the Gene Expression Omnibus by two independent groups. Next, the authors performed bioinformatics analysis in order to find the association of particular miRNAs with mRNAs. Also the H9C2 cell and samples from AMI patients were used for clinical evaluation of the identified miRNAs and their target genes. The English is clear and acceptable.

Experimental design

The authors do not provide row data for qPCR experiments. Also there is no information who was responsible for this experiments (see “Authors’ contributions”).
2.) There is no baseline characteristics of MI patients and control group (sex, age, clinical parameters etc.) The authors do not give results of miRNAs expression in blood samples patients. Whether they were consistent with the results obtained for the tissue cultures?
3.) In case of experiments performed on the cell lines there is a lack of access to methodological details, raw data, number of replicates etc.
4.) Authors state that “miR-139-3p, miR-24-1*, miR-598 and miR-142-3p, have not 259 been linked to the regulation of cardiac performance” what is not entirely true because all of them were mentioned in articles connected with cardiovascular processes:
- miR-139-3p Sci Rep. 2017 Aug 21;7(1):8898.
- miR-24-1 - Circ Heart Fail. 2019 Mar;12(3):e005765.
- miR-598 - Eur J Heart Fail. 2013 Oct;15(10):1138-47.
- miR-142-3p–PloSOne.2017 Jan 26;12(1):e0170842.; Biomed Pharmacother. 2018 Dec;108:1347-1356; J Mol Cell Cardiol. 2019 Aug;133:12-25.

Validity of the findings

A number of previous studies have already shown that plasma miRNAs expression changes in patients in the acute phase of a myocardial infarction. While the association of particular miRNAs with MI was suggested in this work, evidence is lacking in supporting it being a prognostic biomarker.

·

Basic reporting

no comment

Experimental design

no comment

Validity of the findings

no comment

Additional comments

In their manuscript entitled, ‘The biomarkers of key miRNAs and target genes associated with acute myocardial infarction,’ Wang et al. explored miRNA expression in cell culture and clinical samples of myocardial infarction (MI) and controls in order to identify candidate markers for the diagnosis of acute MI

The manuscript seems to attempt to address a need for more sensitive biomarkers for the detection of MI.
However, it does not become clear why there is a need for new biomarkers in the first place. Furthermore, the manuscript is not providing structured information as to exactly which types of experiments were done at which time point and in which sample types. Most importantly, it remains unclear what the overall intention of the manuscript is – discovery of novel circulating biomarkers for MI? discovery of novel miRNA-related signalling pathways?


1) It does not become clear what the intention of the manuscript is. Is it to identify novel circulating biomarkers for a potential clinical application in the diagnosis of acute MI or is it to investigate miRNA pathways on a cardiac cellular level?

2) It does not become clear what was done in the manuscript:
- Lack of information on the type of samples used
- Lack of information on which samples (cell culture, blood (which blood fraction – whole blood? Blood cells?)) were used for which analysis
- Which sample/cell type was “Identification of DE-miRNAs”, “MiRNA – target gene interactions”, “Enrichment analyses of the target genes”, Protein-protein interaction network”, “Biological analysis of the hub genes” performed in?

3) qPCR-based validation was performed in clinical samples in MI patients vs. controls. The patients need to be characterised  baseline characteristics (i.e. age, gender, cardiovascular risk factors, medication). Especially anti-platelet medication is of importance since this alters platelet-derived miRNAs.

4) Immensely important is the application of heparin in patients/individuals before blood was drawn, since heparin is a strong confounder of qPCR-based DNA/RNA detection methods. Were any efforts undertaken to overcome the heparin bias (heparinase treatment?)

5) With qPCR-based RNA quantification, normalisation is of utmost importance. How was the discovery of normalizers carried out and which normalizers were used? How were the qPCR raw data transferred into relative quantity? Delta-delta Cq method?

6) The AUC analyses need to be compared with established biomarkers for AMI (i.e. troponins). Otherwise the results cannot be interpreted properly.

7) In the abstract it does not become clear why there is a need for additional diagnostic biomarker for AMI. It is widely known that cardiac troponins can be detected highly sensitive and at the same time show high specificity. The broad international audience might want to learn about the necessity to explore alternative biomarkers.
In line with this, the authors state in line 60 “…high mortality of AMI is its late diagnosis, which is induced by the lack of highly sensitive and specific biological markers.” In the light of the highly sensitive and specific cardiac troponin biomarkers, this sentence is wrong and needs to be amended; the current ‘problem’ with high sensitive cardiac troponin is rather the high sensitivity which comes along with more patients being evaluated in the emergency department before AMI can be ruled out and thus causing higher socio-economic costs.

8) Furthermore, beginning in the abstract the authors should specify what type of samples were used.

9) As per the above point 4, please specify in the method section how blood was taken, which blood collection tubes were used (RNA-protective reagents such as PAXGene or EDTA etc.?). This is particularly important since upon freezing blood cells will lose their integrity and release their content into the cell free compartment. After thawing this released DNA and or RNA is prone to degradation.

10) The term ‘normal samples’ most likely refers to control samples I suppose? I suggest to use the latter term.

---

## Round 0.2 · accepted · Accept

The Reviewer did not require any further revision.

Reviewer 1 ·

Basic reporting

No comment

Experimental design

No comment

Validity of the findings

No comment